# Neonatal Consumption of Oligosaccharides Greatly Increases L-Cell Density without Significant Consequence for Adult Eating Behavior

**DOI:** 10.3390/nu11091967

**Published:** 2019-08-21

**Authors:** Gwenola Le Dréan, Anne-Lise Pocheron, Hélène Billard, Isabelle Grit, Anthony Pagniez, Patricia Parnet, Eric Chappuis, Malvyne Rolli-Derkinderen, Catherine Michel

**Affiliations:** 1Nantes Université, INRA, UMR1280, PhAN, F-44000 Nantes, France; 2IMAD, F-44000 Nantes, France; 3CRNH-Ouest, F-44000 Nantes, France; 4Olygose, parc Technologique des Rives de l’Oise, F 60280 Venette, France; 5Nantes Université, INSERM, UMR 1235, TENS, F-44000 Nantes, France

**Keywords:** prebiotic, gut-brain, programming, microbiota, L-cell, eating behavior

## Abstract

Oligosaccharides (OS) are commonly added to infant formulas, however, their physiological impact, particularly on adult health programming, is poorly described. In adult animals, OS modify microbiota and stimulate colonic fermentation and enteroendocrine cell (EEC) activity. Since neonatal changes in microbiota and/or EEC density could be long-lasting and EEC-derived peptides do regulate short-term food intake, we hypothesized that neonatal OS consumption could modulate early EECs, with possible consequences for adult eating behavior. Suckling rats were supplemented with fructo-oligosaccharides (FOS), beta-galacto-oligosaccharides/inulin (GOS/In) mix, alpha-galacto-oligosaccharides (αGOS) at 3.2 g/kg, or a control solution (CTL) between postnatal day (PND) 5 and 14/15. Pups were either sacrificed at PND14/15 or weaned at PND21 onto standard chow. The effects on both microbiota and EEC were characterized at PND14/15, and eating behavior at adulthood. Very early OS supplementation drastically impacted the intestinal environment, endocrine lineage proliferation/differentiation particularly in the ileum, and the density of GLP-1 cells and production of satiety-related peptides (GLP-1 and PYY) in the neonatal period. However, it failed to induce any significant lasting changes on intestinal microbiota, enteropeptide secretion or eating behavior later in life. Overall, the results did not demonstrate any OS programming effect on satiety peptides secreted by L-cells or on food consumption, an observation which is a reassuring outlook from a human perspective.

## 1. Introduction

Preventing unhealthy feeding behavior is highly desirable since deleterious eating habits are associated with health problems, including a higher risk of overweight and obesity [1]. Since eating behavior is the result of integrated central and peripheral biological systems that are influenced by genetic, psychological, and environmental factors [2], its optimization is highly complex and requires the full elucidation of the mechanisms that control eating behavior. Central regulation of appetite is mediated by peripheral inputs generated by stomach distension, through signals from the gut epithelium when it senses the availability of nutrients, such as satiety-regulating peptides synthetized and released by enteroendocrine cells (EECs), as well as by long-term energy signals released by adipose tissue and cerebral inputs generated by hedonics and rewards circuits [2,3].

In addition to the evident progress in understanding these interconnections, recent advances include two major findings: first, eating behavior may be programmed very early in life, and second, it could be regulated by intestinal microbiota.

According to the developmental origin of health and disease (DOHaD) theory, adverse early-life conditions may predispose a person to disordered eating [4]. Among the environmental stressors that may have an effect, it is suggested in both animal and human studies that perinatal nutrition could program the appetite (see [5,6] for reviews). In rodents, experiments based on restricting maternal nutrition and/or manipulating litter size have demonstrated that both pre- and post-natal nutrition may alter food intake [7,8,9] and/or food preference [10] in offspring, with subsequent repercussions in adulthood. In humans, although controversial results have been observed concerning the influence of prenatal nutrition on later eating behavior (see [11] for review), some observational evidence suggests that early nutrition/growth affects appetite regulation [12,13,14] and food preference programming, as demonstrated after repeated exposure to new flavors [15].

With regard to the involvement of intestinal microbiota in feeding behavior, although it has been known for several years that fermentation catalyzed by intestinal microbiota stimulates the expression of satiety peptides by EECs [16,17], it is only recently, in connection with the growing appreciation of the role that intestinal microbiota play in regulating host physiology, that this topic has generated renewed interest [18,19]. As pointed out in these reviews, some observations objectively support the involvement of intestinal microbiota in the regulation of feeding behavior. Thus, in ascending order of convincing power, we can quote: (i) the differences observed in microbiota composition or diversity in patients with anorexia nervosa (see [18] for review, [20]); (ii) the fact that feeding behavior differs between germ-free and conventional animals (see [21] for an example), (iii) the ability of certain microbiota modulating agents—e.g., certain prebiotic oligosaccharides [22,23]—to affect feeding behavior, and (iv) the delineation of mechanistic pathways that link microbiota with central and peripheral neuroendocrine systems responsible for feeding behavior, a finding which supports the existence of a causative link. For example, EECs that secrete appetite-regulating peptides can be mentioned since they have a large diversity of receptors enabling them to sense microbial inputs such as fermentation-derived short chain fatty acids (SCFA), secondary biliary salts or pathogen-associated molecular patterns [see 18 for review].

Reconciliating these two emerging issues related to the regulation of feeding behavior, i.e., its possible programming in early life and its control by intestinal microbiota, we hypothesized that early modifications to microbiota may program adult feeding behavior. This programming could stem from either the programming of intestinal microbiota (e.g., [24]) or the early impacts of microbiotal changes with long-lasting consequences for the peripheral neuroendocrine systems that control adult feeding behavior and/or the central sensing of it. In this respect, it is worth mentioning the ability of microbiota-modulating agents to affect the hypothalamic expression of neurogenic factor (BDNF) during the neonatal stage [25], and the potential programmable character of both the EECs [26] and the vagal sensitivity [9]. In addition, the putative ability of the gut microbiota to act through epigenetic mechanisms (see [27] for review) as well as the ability of the microbiota presence [28] and certain microbiota modulating agents [e.g., in the case of prebiotics [29,30]) to modulate some behaviors in adults mice can be cited, assuming that they are transposable in the neonatal period.

Using the rat as a model, we therefore evaluated whether neonatal modulation of the microbiota induced by prebiotics could program eating behavior and the secretion of gastrointestinal peptides in adulthood. We first verified that the presuppositions underlying our hypothesis were present in our case, by investigating the immediate impact of the neonatal prebiotic supplementation on both the intestinal microbiota and the maturation and functioning of EEC in suckled rats. We decided to use indigestible oligosaccharides (OS) to modify the intestinal microbiota of the neonatal rats for two reasons: first, OS are recognized as intensively-fermented prebiotics [31], which are also operant in neonatal rats [24] and infants (see [32] for review) and have been shown to stimulate EEC proliferation and activity in adult animals [33,34]; second, they represent relevant nutrients in neonatal nutrition since they are commonly added to infant formula to better mimic maternal milk [35].

## 2. Materials and Methods

### 2.1. Ethics Statement

All experiments were conducted in accordance with the European Union Directive on the protection of animals used for scientific purposes (2010/63/EU). The protocols were approved by the Ethics Committee for Animal Experiments for the Pays de la Loire region (France) and the French Ministry of Research (APAFIS#3652-20 160 1 1910192893 v3). The animal facility is registered by the French Veterinary Department as A44276.

### 2.2. Animal Experiment

Primiparous female Sprague-Dawley rats (*n* = 16) were obtained on day one of gestation (G1) from Janvier-labs (le Genest Saint Isle, France) and housed individually (22 ± 2 °C, 12:12-h light/dark cycle) with free access to water and chow (A03, Safe Diet, Augy, France). At birth, 8 litters were culled to 8 male pups per mother with systematic cross fostering as previously described [24]. From day 5 to day 14/15 of life (PND5 to PND14/15), the pups were given various solutions of FOS, GOS/In mix (9:1), αGOS or a mix of the monomers present in the OS solutions (Table 1) by oral gavage. These OS were selected either because they are already used in infant formula (GOS/In, FOS [35]) or because they constitute a new source of OS, the physiological properties of which are to be characterized (αGOS). Two pups from each litter were given one of the 4 solutions daily.

The pups were weighed daily and the administered volume was adapted to body weight to reach 3.2 g/kg in order to approximate the dosage actually consumed by babies fed with prebiotic enriched formula, taking into account both the difference in metabolic rate between rats and humans and the true prebiotic content of infant formula [35].

Four of the 8 litters were used for our main objective, i.e., to assess eating behavior programming: rats from these 4 litters (*n* = 8 per group) were weaned at PND21 onto standard chow (A03, Safe Diet, Augy, France) in individual cages until PND124/126, when they were sacrificed by decapitation after induction of deep anesthesia (isoflurane/O_2_, 5 L·min^−1^). During the follow-up, food consumption was measured 3 times a week. Rats from the 4 remaining litters (*n* = 8 per group except for FOS where *n* = 7 as explained below) were sacrificed at PND14/15 by the method described above to investigate the immediate impact of the neonatal prebiotic supplementation on both intestinal microbiota and the maturation and functioning of EECs.

This experimental set-up was designed to form 8 supplemented males, originating from 4 different litters, per group at each studied age. Due to the death of one of the pups during the supplementation period (this pup was then replaced by an untreated one to equilibrate the litter size), the number of pups in the FOS group at PND14/15 was reduced to 7. These values are maximum numbers that are not always found in each of the analyses (see the illustration legends). This stemed from either physiological reasons (e.g., 2 animals did not eat at all during the fasting-refeeding test), or because of quality requirements (e.g., reliable data from in physiological cages could only be obtained for *n* = 7 in CTL and GOS/In groups; *n* = 6 in FOS group and *n* = 5 in αGOS group), or statistical inconsistency (e.g., outliers identified by the Dixon’s Q test were excluded in RT-qPCR analysis as well as food/beverage consumption follow-ups), or technical problems (e.g., accidental spillage of supernatant before analysis of bacterial end-products or sequencing failure during 16S rDNA analysis or poor quality of some tissue sections in the case of immunochemical analysis). Nevertheless, in all analyses, the 4 different litters were always represented.

### 2.3. Tissue Collection

Under anesthesia, intracardiac blood was collected in a tube containing EDTA (Microtubes 1.3 mL K3E, Sarstedt MG & Co, Marnay, France) and plasma collected after centrifugation 2000× *g*, 15 min, 4 °C) was frozen at −20 °C for further analysis. The contents of the most distal 15 cm of the ileum were harvested by flushing, using 1 mL of Hanks’ Balanced Salt Solution (HBSS, Thermo Fisher Scientific, St-Herblain, France), and the cecocolonic (PND14/15) or cecal (PND124/126) content was collected, weighed, mixed with 5-fold or 2-fold their volume of sterile water (PND14/15 and PND124/126, respectively). After complete homogenization, these cecocolonic/cecal suspensions were centrifuged 7800× *g*, 20 min, 4 °C) then both supernatants and pellets were frozen at −20 °C for analysis of the fermentation end-products (SCFA and lactate) and microbiota, respectively. Intestinal tissues (ileum and proximal colon) were rapidly collected and frozen in liquid nitrogen for RNA analysis. Additional tissue samples were fixed in 4% paraformaldehyde for immunofluorescence analysis.

### 2.4. Eating Behavior

#### 2.4.1. Meal Pattern

Between PND74 and PND99, eating behavior was analyzed in physiological cages (Phecomb cages, Bioseb, Vitrol, France) as previously described [8]. Briefly, the rats were housed individually and following 24 h of acclimatization to the cage and refilling with fresh food between 9.00 a.m. and 11.00 a.m., data were recorded every 5 s over a 20-h period. Due to the intervention during the diurnal phase, the analysis was reduced to 8 h whereas the nocturnal phase was 12 h. The exact feeding pattern was defined with a minimal size of 0.1 g, a minimum duration of 10 s and a minimum inter-meal interval of 10 min. Events such as large vibrations (contact with the feed tray without eating) were filtered by the Phecomb system monitoring software (Compulse v1.1.01). The reliability percentage of the quality signal was calculated by the software and only experiments with a percentage >80% were used. Meal parameters extracted from Compulse software included number of meals, meal size and duration, inter-meal intervals and satiety ratio.

#### 2.4.2. Taste Preference

Preference for sweet taste was measured at PND110 using the bottle test experiment [36]. After a two-day habituation to the presence of two bottles in their own cages, the animals had the choice of the two bottles, one containing tap water and the other 0.05% saccharin (Sigma-Aldrich, St. Quentin Fallavier, France). Drink intake was measured daily for three days. The position of the two bottles was reversed each day to prevent position preference bias. The sweet preference score was calculated as the ratio between the volume of saccharin solution consumed and the total drink intake in 24 h, then multiplied by 100. Preference was defined as a percentage greater than 50.

### 2.5. Fasting-Refeeding Test, Kinetics of GLP-1 and PPY Release and Response to Glucose

At PND105, a 4 h kinetic of GLP-1 and PYY release in plasma was carried out. Rats were not fed for 16 h to induce hunger and then fed for 20 min with a calibrated quantity of chow (A03, safe Diet). Food intake was weighed at the end of the 20 min period. Any crumbs that fell in the cage were weighed and deducted from the food intake. Blood samples were collected from the tail vein in tubes containing EDTA (Microvette CB300 EDTA 3K, Sarstedt, Marnay, France) at 0 (15 min before refeeding), 30, 60, 120, and 180 min after the beginning of the meal.

At PND124/126, the rats were not fed for 16 h, and 2 h before being sacrificed they were given an oral bolus of glucose (2 kg/kg BW) in order to challenge glucose sensing in GLP-1/PYY-producing EECs.

### 2.6. Plasma Gastrointestinal Peptides

The plasma concentration of total GLP-1 and total PYY was assayed by the ELISA technique using kits from Millipore (Merck- Millipore, Molsheim, France) and Phoenix Pharmaceutical (Phoenix France S.A.S, Strasbourg, France), respectively. 

### 2.7. Fermentation End-Products

Ileal and cecal supernatants were centrifuged 8000× *g*, 20 min, 4 °C, diluted (1/10) with 0.5 M oxalic acid and SCFA (acetate, propionate, butyrate, isobutyrate, valerate and isovalerate) were analyzed by gas chromatography as previously described [37]. The D-and L-lactates were measured in the supernatants after heating to 80 °C for 20 min with a D/L-lactic acid enzymatic kit following the manufacturer’s instructions (Biosentec, Toulouse, France). 

### 2.8. Immunochemistry

Tissue sections (4–5 μm) of fixed ileum and proximal colon were double-stained with a goat polyclonal antibody raised against GLP-1 diluted at 1/200 (Santa Cruz Biotechnology Inc, Santa Cruz, USA) and a rabbit anti-chromograninA (chrgA, diluted at 1/1000 (ImmunoStar Inc, Hudson, USA), followed by incubation with anti-goat and anti-rabbit fluorescent secondary antibodies (1/1000). Nuclei were counterstained with 4′,6-Diamidine-2′-phenylindole dihydrochloride (DAPI, Sigma-Aldrich, St. Quentin Fallavier, France). Tissues sections were mounted in Prolong Gold anti-fading medium (Molecular Probes, Thermo Scientific, Courtaboeuf, France). Three sections per sample were analyzed with a Nanozoomer (×20) (Hamamatsu Photonics France, Massy, France). The number of fluorescent cells along the crypt-villus axis unit was counted twice by a blind operator, using the NDP view software (Hamamatsu, Photonics France, Massy, France). A total of 40 to 60 crypt-villus units per section were counted.

### 2.9. Quantitative Real-Time PCR

Total RNA extraction from the ileum and colon was carried out using a QIAamp RNA Blood Mini kit (Qiagen, Courtaboeuf, France) following the manufacturer’s instructions. Two micrograms of RNA were reverse-transcribed using M-MLV reverse transcriptase (Promega, Charbonnières-les-Bains, France). Five microliters of 1/40 dilution of cDNA solution were subjected to RT-qPCR in a Bio-Rad iCycler iQ system (Biorad, Marnes-la-Coquette, France) using a qPCR SYBR Green Eurobiogreen^®^Mix (Eurobio, Les Ulis, France). The quantitative PCR consisted of 40 cycles, 15 s at 95 °C, 15 s at 60 °C and 15 s at 72 °C each. Primers sequences are shown in Appendix A. For quantification of Neurog3, rat PrimePCR^TM^ SYBR^®^GreenAssay Neurog3 (Biorad, Marnes-la-Coquette, France) was used. Relative mRNA quantification was expressed using the 2^−∆∆Cq^ method with actin gene as a reference. 

### 2.10. Bacterial 16S rDNA Sequencing of Cecal Contents

DNA was extracted from pellets of ceco-colonic content (max 250 mg) using the QIAamp Fast DNA Stool Mini kit (Qiagen, Courtaboeuf, France) after enzymatic and mechanical disruptions as described previously [37] except that homogenization was carried out at 7800 rpm for 3 × 20 s intervals with 20 s rest between each interval in a Precellys^®^ “evolution” bead-beater (Bertin, Montigny-le-Bretonneux France). The V4 hyper-variable region of the 16S rDNA gene was amplified from the DNA extracts during the first PCR step using composite primers (5’-CTTTCCCTACACGACGCTCTTCCGATCTGTGYCAGCMGCCGCGGTAA-3′ and 5’-GGAGTTCAGACGTGTGCTCTTCCGATCTGGACTACHVGGGTWTCTAAT-3′) based on the primers adapted from Caporaso et al. (i.e., 515F and 806R) [38]. Amplicons were purified using a PP201 PCR Purification Kit (Jena Bioscience, Jena, Germany). Paired-end sequencing was performed on a HiSeq 2500 System (Illumina, San Diego, CA, USA) with v3 reagents, producing 250 bp reads per end, following the manufacturer’s instructions by the GeT+-PlaGe platform (INRA, Toulouse, France). The 16S rDNA raw sequences were analyzed with FROGS v2 pipeline (http://frogs.toulouse.inra.fr/) [39]. After de-multiplexing, quality filtering and chimera removing, the taxonomic assignments were conducted for OTUs with abundance >0.005% with Blast using Silva 128 database containing sequences with a pintail score at 80 to determine the bacterial compositions. FROGSSTAT Phyloseq tools were used to normalize raw abundances by rarefaction and to calculate alpha and beta diversity indices.

### 2.11. Statistical Analysis

Statistical analyses were carried out using GraphPad Prism 6 software (GraphPad Software Inc., San Diego, USA) or R (librairies “stats v3.5.1” and “corrplot v0.84”, [40]). Differences between treatments were searched using one-way ANOVA followed by Tukey’s multiple comparison tests for most data, with the exception of growth and food consumption data which were subjected to multiple t-tests with correction for multiple comparison using the Holm-Sidak method. Sweet taste preference test was analyzed by the one sample t-test to compare to compare data against the 50% (no preference) value. A *p* value < 0.05 was considered statistically significant.

## 3. Results

### 3.1. Neonatal OS Supplementation Did Not Substantially Affect Rat Growth

Both FOS and αGOS supplementation was associated with a significant transitory reduction of pup growth in the first days of intervention (PND7 to PND10 and PND6 to PND8 respectively,Appendix A). When compared with body weights from the CTL group, the differences observed were only 9.1 to 11.5% and did not significantly affect the cumulative weight gains measured either from birth until the end of supplementation or for the whole lactation period (Table 2).

No significant differences in bodyweight were observed between groups after weaning (Appendix A).

### 3.2. Neonatal OS Supplementation Exerted a Marked Immediate Impact on Intestinal Environment

#### 3.2.1. OS Supplementation Modified Both Composition and Activity of Neonatal Intestinal Microbiota

Following 16S rDNA sequencing, no significant differences were noticed in raw sequence numbers between cecocolonic samples collected at PND14/15 (355,245 ± 10,367, 30,306 ± 13,817, 40,275 ± 18,343 and 31,808 ± 10,101 for CTL, FOS, GOS/In and αGOS, respectively) or in percentages of sequences kept after quality filtering (83.8 ± 4.0, 76.4 ± 18.4, 83.9 ± 4.2, and 81.7 ± 7.1). The cecocolonic contents of animals supplemented with OS exhibited similar reductions in richness (*p* < 0.001) compared with CTL animals (Chao1 values: 66.2 ± 21.0, 72.9 ± 28.1, and 73.9 ± 35.3 for FOS, GOS/In and αGOS, respectively *versus* 180.0 ± 35.7 for CTL). The cluster dendrogram generated using weighed UniFrac metric which illustrates beta or between-sample diversity, highlighted an obvious dissimilarity between the microbiotas of the OS-supplemented animals and those of animals from the CTL group (Figure 1) but did not reveal any effect of the nature of the OS.

When considering bacterial families occurring at more than 0.01% of the total sample abundances (Table 3), the OS impact was typified by significant decreases in Lactobacillaceae, Bacteroidales S24-7 group, Prevotellaceae, Streptococcaceae, Peptococcaceae, Coriobacteriaceae, Aerococcaceae, Family XIII, and Rikenellaceae. In addition, OS supplementation decreased Ruminococcaceae abundance but this impact only reached statistical significance for FOS and αGOS. These decreases in relative abundance were differently compensated according to the OS: increases in Bifidobacteriaceae reached statistical significance following FOS and αGOS supplementations, Enterobacteriaceae increased following αGOS supplementation and Lachnospiraceae increased following GOS/In supplementation.

Significant differences between OS were scarce and only occurred between GOS/In and αGOS in their impact on Lachnospiraceae (Table 3).

Concurring with these compositional changes, the 10-day supplementation greatly affected fermentation end-product concentrations in both ileal and colonic contents at PND14/15. 

In the ileum, lactate concentration was below the detection limit (0.22 mM) in all animals, and the concentration of acetate—the sole SCFA present at this age in this intestinal segment—was significantly increased (*p* < 0.005) through FOS supplementation (6.9 ± 3.6 mM) compared to CTL (0.3 ± 0.4 mM), GOS/In (1.6 ± 2.0 mM) and αGOS (0.6 ± 0.8 mM).

In the cecum, the concentration of total end products increased in all OS groups compared to CTL (Figure 2). This was mainly due to an increase in SCFA concentration, which only reached statistical significance in the case of FOS and also an increase in lactate concentration in the case of αGOS.

Increases in total SCFA reflected acetate increases which were significant for both FOS and GOS/In groups, and paralleled significant decreases in pH values (Table 4). In addition, OS supplementation shifted microbiotal activity, as evidenced by significant changes in the relative proportions of acetate (93.8 ± 4.6, 93.1 ± 4.1, and 95.4 ± 2.9% for FOS, GOS/In and αGOS, respectively *versus* 86.3 ± 4.5% for CTL) and propionate (5.4 ± 4.6, 5.2 ± 3.4, and 3.6 ± 2.9% for FOS, GOS/In and αGOS, respectively versus 10.7 ± 3.0% for CTL). Concentration and relative proportions of butyrate—which is scarcely produced in the neonatal stage—were not affected significantly by supplementation.

#### 3.2.2. OS Supplementation Modified both Differentiation and Activity of the Neonatal EEC

In the ileum, a profound effect on the enteroendocrine lineage was induced by neonatal OS supplementation, as revealed by a significant decrease in *Neurog3* expression in the OS groups compared to CTL, whereas, an early expressed marker in the commitment secretory lineage (*Atoh1*) was not affected significantly (Figure 3). The related expression of genes specifically implied in the differentiation of EECs (*Pax4* and *Pax6*) decreased significantly in OS supplemented groups compared to CTL, whereas expression of *Foxa1* did not vary between the groups. Similar to *Pax4* and *Pax6*, *Neurod1* expression decreased in OS groups compared with CTL, but this did not reach statistical significance for FOS. Regarding the expression of gene coding for peptides produced by mature L-cells, *Pyy* increased significantly in OS groups compared to CTL. At the same time, despite a 2-fold increase in *Gcg* expression in the OS groups compared to CTL, this effect was not statistically significant due to the widely varying expression between samples.

In the proximal colon, the impact of OS supplementation was much more moderate and their only significant effect was a decrease in the expression of *Pax4* (Appendix A).

Along with this profound remodeling in the expression of markers of L-cell differentiation, the number of GLP-1/ChgrA positive cells, i.e., mature EECs, was higher in the ileum of pups from OS groups compared to CTL but only reached statistical significance for villi (Figure 4A–C).

In agreement with this rise in the number of mature enteroendocrine cell (EEC), plasma concentrations of GLP-1 (Figure 5A) and PYY (Figure 5B) were significantly increased by all the neonatal OS supplementations, as compared with CTL.

Significant positive associations between plasma concentrations of GLP-1 and PYY and the ileal expression of their respective genes were evidenced (Figure 6A). Conversely, these plasma concentrations as well as the density of GLP-1 secreting cells, were inversely correlated with expressions of *Neurog3*, *Neurod1*, *Pax4*, and *Pax6*. With respect to associations between microbiota and EEC descriptors (Figure 6B), only some of the differentiating factors (*Pax4, Neurod1, Pax6* and *Neurog3*) exhibited significant positive correlations with the abundance of some bacterial families corresponding to those the abundance of which was significantly reduced by OS, except for Prevotellaceae. For these factors, the sole negative correlation was that between *Neurod1* and abundance of Clostridiaceae.1. Conversely, the PYY and GLP-1 plasmatic concentrations, EEC densities and *Pyy* expression, but not *Gcg* expression, were negatively correlated with the same families including Prevotellaceae.

Overall, these results indicate that OS supplementation profoundly modulates neonatal microbiota in terms of both its composition and its fermentative activity, with repercussions not only in the cecocolon but also, as exemplified with FOS, in the ileum. An increased density of ileal L-EECs and their secreted anorectic hormones, GLP-1 and PYY, were observed and unexpectedly the expression of transcription factors beyond the stage of secretory cell engagement (Atoh1) was inhibited at the same time. Whether this strong impact of early OS supplementation on satiety peptide-related EECs could last into later life and affect eating behavior was investigated further.

### 3.3. Neonatal OS Supplementation Had No Significant Long-Term Consequences

#### 3.3.1. Neonatal OS Supplementation Did Not Significantly Program Enteropeptide Production or Eating Behavior in Adulthood

To investigate the long-term effect of neonatal supplementation of OS on nutrient sensing in EECs, once pups reached adulthood, we studied the release of GLP-1 and PYY in response to both a 20-min test meal (PND 74/76) and an oral bolus of glucose (PND 124/126) after 16 h of fasting. 

No significant differences were observed between groups in the amount of food consumed during the 20-min test meal (Figure 7A). In response to this meal, the plasma concentration of GLP-1 increased immediately after refeeding and returned to pre-prandial level 120 and 180 min later (Figure 7B). The total amount of GLP-1 secreted during this period, quantified by AUC, did not differ significantly between the groups (Figure 7C). PYY secretion did not show any postprandial peak or significant differences between the groups (data not shown).

Similarly, at PND 124/126, plasma concentrations of GLP-1 (CTL: 34.4 ± 13.5; GOS/In: 38.6 ± 28.6; αGOS: 28.9 ± 10.0 and FOS: 37.9 ± 20.6 pM) and PYY (CTL: 84.7 ± 4.0; GOS/In: 88.7 ± 7.8; αGOS: 91.4 ± 7.3 and FOS: 91.5 ± 5.8 pM) measured 2h after an oral bolus of glucose did not show any significant difference between groups. 

To investigate the long-term effect of a neonatal supplementation of OS on subsequent eating behavior, we followed up the food consumption from weaning to adulthood, performed a refined analysis of feeding pattern using physiological cages from PND75 to PND100 and assessed the preference for sugar taste between PND109 and PND111.

The analysis of food consumption during development, expressed per Kg of body weight to allow for strict comparison, only revealed a single significant difference which occurred at PND32 between animals from the FOS and CTL groups (Figure 8), an observation which indicates that neonatal supplementation with OS did not greatly influence the subsequent food intake in our experimental conditions.

This absence of effect on daily food consumption was confirmed by a detailed analysis of food consumption: we observed no significant difference in meal patterns among the groups (food intake, food intake per meal, number and duration of meals, latency to eat the first nocturnal meal, satiety ratio and ingestion rate), whatever the period of measurement (total 20 h period of measurement, diurnal period (8 h) or nocturnal period (12 h) (Figure 9 andAppendix A). 

In the sweet taste preference test, there was no significant difference between groups in terms of the consumption of saccharin solution expressed as a percentage of daily beverage intake, regardless of the day of testing (Figure 10). Strikingly, the preference for sweet taste for the GOS/In group did not reach statistical significance on the first day of the test, in contrast to the FOS and αGOS groups. However, this preference did not persist on day 2, contrary to what was observed for the CTL group. This suggests that neonatal supplementation with OS slightly reduced the persistence of sweet preference in adulthood.

#### 3.3.2. Neonatal OS Supplementation Did Not Significantly Program Adult Intestinal Microbiota

At adult age (PND 124/126), no significant differences were observed between treatments with respect to the raw number of sequences obtained, percentages of sequences kept after quality filtering, or alpha-diversity indexes (data not shown). Similarly, β-diversity analysis (Appendix A), principal component analysis on OTU abundances (Data not shown) and comparisons of the cumulated relative abundances at family level (Figure 11) failed to show any significant difference between cecal samples with respect to neonatal supplementation. Finally, neither ileal nor cecal concentrations of SCFA showed significant differences between the groups (Appendix A).

Overall, these data did not reveal that neonatal OS supplementation had any programming effect on adult microbiota.

## 4. Discussion

Considering that the regulation of feeding behavior could be programmed from the beginning of life and controlled by intestinal microbiota, we hypothesized that modifications to the neonatal microbiota could program adult feeding behavior. We therefore checked the ability of prebiotic-induced intestinal microbiota modulations to affect the maturation and functioning of L-EECs in suckled male rats, then assessed whether this resulted in delayed alterations in eating behavior and the secretion of GI peptides in adulthood. The observed effects are specifically attributable to OS since we adjusted the compositions of the administered solutions by taking into account the digestible sugar contents of commercial OS sources. In this study, we show that neonatal supplementation with 3 different OS strongly impacts cecocolonic microbiota, GLP-1 cell density in the ileum, and the production of satiety-related peptides during the neonatal period, but does not induce any significant enduring effect in adulthood on either eating behavior or gut peptide secretion. 

The validity of this statement is obviously limited to our operating conditions which represent both strengths and limitations for our study. 

Limitations include the fact that we only studied males in order to avoid the already described fluctuations in food intake throughout the estrous cycles [41], and did not characterize every components of eating behaviour such as motivation. However, we believe that the numerous components investigated allow for consideration of both its homeostatic and hedonic elements. We did not investigate immediate impact of OS supplementations on feeding behavior to avoid the recognized stress induced in pups by separation from the mother which would have been required to quantify milk intake either by gravimetric [42] or deuterated water turnover methods [43]. We did consider moreover whether neonatal prebiotic supplementation having an impact on the pups’ eating behavior was beyond the scope of the programming of adult eating behavior. Nevertheless, we reported a transitory reduction in BW gain in FOS and GOS/In groups between PND6 and PND10 which suggests that the reducing impact of OS on food intake may also operate in the neonatal period. 

Inversely, our study has three major advantages: the combination of hormonal, behavioral and microbiological analyzes; the minimizing of the influence of lactating mother influence by supplementing pups from the same litters with the different OS, and finally the use of OS doses comparable to those actually consumed by toddlers.

### 4.1. Neonatal OS Supplementation Affected Intestinal Microbiota Despite Its Immaturity

Corroborating our previous findings based on a non-exhaustive analysis of the microbiota [24], and in concurrence with several in vivo and in vitro studies investigating the impact in adulthood of OS (including those of the αGOS [44]) on intestinal microbiota, in humans and animals (e.g., [45,46]) and in human infants (see [32] for review), all the oligosaccharides used here dramatically affected neonatal microbiota in rat pups. This confirms that the prebiotic properties previously demonstrated in adult rodents (e.g., [30,45,46]), also operate in neonatal pups despite the immaturity of the microbiota at this stage of development [37]. 

In addition to these changes in composition and the reduction in microbiota richness, our neonatal OS supplementations also modified the activity of the microbiota by stimulating the production of acetate and lactate at the expense of that of propionate. This decrease in propionate concentration stands out from what is observed in adult rats, for which GOS and FOS are frequently reported as being particularly stimulating for propionate and/or butyrate production (e.g., [34,47]) and could be related to the known progressive maturation of the microbiotal capacity to synthesis the different SCFAs in neonates [37,48]. The production of butyrate is therefore barely detectable before the day 16 of life in rats [37]. In any case, the neonatal OS supplementation we performed resulted in microbiota that differed greatly from that of unsupplemented animals, an observation which was a prerequisite for investigating the ability of neonatal microbiota modulation to program adult eating behavior or gut peptide response.

### 4.2. OS Supplementation May Stimulate Ileal EECs to Produce GLP-1/PYY While Acting in Feedback on Endocrine Precursors

Our results showed that neonatal OS supplementation had immediate effects on mature ileal GLP-1-cells by increasing the density in villi and the mRNA expression of *Gcg* and *PYY* leading to enhanced plasma concentrations in these two anorectic peptides. These new observations in neonatal rats are consistent with those reported in adult rats for FOS and GOS/In [33,34,49,50,51] and are, to our knowledge, reported here for the first time for αGOS. In one of these previous studies, this increased production of GLP-1 was related to a higher differentiation of *Neurog3*-expressing EEC progenitors into L-cells in the colon [50]. Here, we demonstrate a drastic down-regulation of endocrine lineage-devoted genes during OS supplementation, mainly in the ileum. This unexpected result is difficult to reconcile as an effect of OS on early endocrine precursors leading to the production of more L-cell subtypes.

Neurog3 marks the endocrine progenitors and is essential for generating new EECs [52]. Post-neurog3 differentiation and maturation of EECs is controlled by dynamics in transcriptional factors such *Neurod1*, *Pax4* and *Pax6* and many others (*Arx*, *Pdx1*, *Foxa1* and *Foxa2*). The hierarchy of these events is still poorly understood [53] and the extrinsic factors that may interplay remain largely unknown. For this study, the well-known effect of OS prebiotics in stimulating L-cells cannot simply be explained by the impact on endocrine precursors, as suggested in the above-mentioned study [50]. Since we know that *Neurog3* expression is restricted to immature proliferative cells, the decreased Neurog3 expression we observed in the ileum may instead reflect a feedback regulation to limit new EEC generation in response to OS supplementation. A similar observation (decreased duodenal Neurog3 and increased EEC density) was reported in a model of maternal deprivation [26]. These data and our own suggest that the postnatal environment affects the differentiation of EEC precursors but not the proliferation of progenitors, leading to increased EEC density. High levels of circulating GLP-1 have been previously attributed to the increased number of ileal L-cells in *Gcgr*-deleted mice, and this effect involved up-regulation of post-*neurog3* transcription factors, affecting the proliferation of L-cells precursors [54]. Here, the expression of these factors, i.e., *Neurod1*, *Pax4* and *Pax6*, was reduced in OS-supplemented groups with high circulating levels of GLP-1, suggesting a different mechanism in the increased density of L-cells. In this respect, it should be noted that although EECs are still classified according to their major/unique hormone product (as for example GLP-1 for L-cells), it is now acknowledged that EECs are multihormonal [53,55]. In particular, more recent data has demonstrated that mature differentiated EECs display hormonal plasticity, allowing them to change their hormonal products in response to extrinsic factors, such as bone morphogenic proteins (BMP) during their migration along the crypt-villus axis [56,57]. Thus, the increased L-cell density observed here may be the result of the direct effect of OS on this plasticity to produce more GLP-1, independently of early markers of EEC proliferation and differentiation. Interestingly, in this study, the production of CCK—a key early-satiety peptide—was not affected by the OS supplementation at PND14/15 (data not shown) reinforcing the specificity of the effect of OS on EECs in producing GLP-1 and PYY in a segment of gut where CCK is not predominantly produced. How OS can modulate both the identity of EEC subtypes and/or the expression of GI peptides by acting on extrinsic factors (such as villus-produced BMP) needs further investigation. 

### 4.3. What are the Putative Mediators of the Massive Effect of Neonatal OS Supplementation on Ileal L-Cells?

Identification of the small intestine rather than the colon as a privileged site for the action of OS on transcriptional activity has been previously reported in studies involving adult animals [58,59]. Conventionalization of germ-free mice led to similar observations (e.g., [60]). However, a nutritional modulation by OS supplementation may have a different impact on ileal epithelium compared to the absence or presence of microbiota. For example, in the Arora’s study [60], conventionalization of germ-free mice led to the down-regulation of GLP-1 secreting vesicle process in L-cells, whereas we observed an increase in GLP-1 and PYY production. These contrasting results may stem from either inter-individual variability, or more likely the great differences in age between the animals studied. Nevertheless, our data raise the question of how OS modulation of microbiota could act on ileal L-cells. The well-known capacity of SCFA (mainly butyrate but also propionate or even, non-consensually, acetate) to stimulate PYY and/or GLP-1 production [61,62,63] seems inconsistent with our observation of an OS-impact mainly localized in the ileum, within the context of no propionate/butyrate synthesis.

Others potential mechanisms include acidification of the luminal milieu or changes in the pathogen-associated molecular patterns (PAMPs). Zhou et al. [61] showed that changes in pH from 7.5 to 6.5 induce *per se* an increase in *Gcg* expression by STC-1 cells in vitro. Apart from this, it is known that EECs have receptors for PAMPs (i.e., Toll-like receptors) (see [18] for review). This is of particular interest since it has been demonstrated that some bacterial strains elicit GLP-1 secretion through signaling agents of the Toll-like receptor system, as illustrated by the fact that a MyD88 blockade triggers GLP-1 secretion induced by bacteria [64].

### 4.4. OS impact on Eating Behavior, Usually Observed Simultaneously with Their Consumption, Does Not Seem to Be Programmable

Despite a certain disparity in the literature, possibly related to the heterogeneity in dosage or methodology, several studies have reported the beneficial effect of OS prebiotics—mainly fructans but also αGOS, on the eating habits of healthy adults [65,66] or overweight adults [67,68], such as feelings of reduced hunger, increased satiety or reduced energy consumption. Note that the existing literature does not establish whether this is also true in infants, who are frequently given prebiotic supplements. In concurrence with human data, decreased food/energy intake has been evidenced in adult rodents supplemented with fructans [49,51] or βGOS [34]. In both models, these effects have been related to SCFA production by colonic bacteria during OS supplementation. For each of the 3 main SCFAs, i.e., acetate, propionate and butyrate, it has been demonstrated that they reduce energy intake, particularly in rodent models of diet-induced obesity [69,70,71], although conflicting results are reported [72], probably dependent on the mode (orogastric [71,72], intraperitoneal [70], intracerebroventricular [70], colonic delivery via fermentable fibers [69,71], etc.) and duration (acute [69,70] vs. chronic [69,72]) of SCFA or SCFA precursors administration. In humans, this hypothesis has been substantiated for both acetate and propionate by numerous studies focusing on appetite-related parameters (see [73] for review) as well as observations of reduced hedonic response to high-energy foods regulated in striatum [74] or reduced energy intake following the administration of propionate precursors in overweight adults [75]. How these SCFA regulate appetite directly at hypothalamic level [70] or via a vagal-dependent mechanism [71,72], whether or not implicating an enhanced intestinal satiety peptide (GLP-1 and PYY) secretion following SCFA interaction with FFAR receptors on L-cells is still a matter of experimental research in animal models and clinical trials in humans [18].

Since the perinatal environment [6,9,24,26] appears to have a long-lasting impact on each of the microbiota-EEC-brain axis actors, we had assumed that early modulation of the microbiota associated with changes in EECs could program eating behavior, a hypothesis which has remained unexplored until now. However, this hypothesis could not be corroborated in this study as adult feeding behavior did not seem to be significantly affected by early supplementation with OS, which nonetheless increased total SCFA, along with increased release of GLP-1 and PYY and L-cell density at the end of supplementation. This lack of eating behavior programming indicates that none of the presupposed events (i.e., programming of EEC or vagal sensitivity and/or microbiota programming) occurred under the test conditions. In fact, no difference in the expression of *c-Fos* was observed in the nucleus of the solitary tract in the rat’s brainstem 2 h after administering a bolus of glucose in adult rats (data not shown). It therefore seems that depending on the nature and intensity of the perinatal stressor (maternal protein restriction [9], maternal deprivation [26] or postnatal modulation of microbiota by OS) the long-lasting impact is not systematic. For the microbiota, the lack of programming could be related to an inadequacy in the timing for applying the modulation, as discussed below.

### 4.5. Is Programming of the Microbiota Subject to Particular Timing?

In this study, we did not observe any programming effect of neonatal OS supplementation on adult microbiota. This result is in line with what we had previously observed for FOS [24] but contradicts the small-scale programming found after neonatal supplementation with GOS/In in this same study. This discrepancy may result from the difference in methods used to analyze the composition of the microbiota, even if it is counterintuitive, since the 16S rDNA sequencing used here is more exhaustive than the qPCR used previously. As this impact was minor, it may also not have been possible to reproduce under our new experimental conditions, i.e., a new batch of animals, a different room at our animal facility, or even a slight difference in the composition of the semi-purified diets we used, since all these parameters are known to affect the microbiota of laboratory animals (see [76] for review).

The disappearance of this nonetheless drastic effect in our animals at the end of supplementation raises the question of what is the most favorable period for sustainable modulation of the composition of the microbiota. In our experimental protocol, prebiotic supplementation was applied for a short postnatal period and ended before the onset of solid food consumption, whereas studies reporting programming effects for early supplementation with OS on the subsequent composition of the microbiota were based on longer-term supplementation, ranging from the prenatal period (i.e., supplementation of gestating mothers) to complete weaning and even beyond [77,78]. Whether the supplementation we applied was either not early enough, not late enough or not for a long enough time is difficult to establish on the sole basis of this comparison. However, in a study by Fugiwara et al. [77], a difference in adult microbiota composition was observed only in mice offspring that were supplemented with FOS beyond weaning. Whether this was also true in the Le Bourgot et al. study [78] cannot be evaluated since all piglets were supplemented with FOS for a few weeks after weaning. From this, it can be assumed that to be lastingly effective, prebiotics must be able to exert their microbiotal effect after full weaning, thereby controlling the impact of new bacterial sources and changes in dynamics of bacterial populations that result from the switch from maternal milk to solid food. Such a switch has been associated with dramatic changes in microbiota composition and activity both in humans [79] and rats [37]. This hypothesis would explain why the early-life events that are known to affect neonatal microbiota composition (i.e., birth mode, infant feeding etc.) are not associated with significant variations in adult microbiota composition [80], but strict comparisons between the time windows for supplementation are required for this to be validated.

### 4.6. All OS Studied Performed Similarly Despite Differences in Their Chemical Characteristics

In our study, the 3 OS studied led to comparable results in terms of both microbiotal impact and physiological repercussions. With regard to microbiotal changes, the observed modifications, in particular the acidification of the contents, the less diversified production of SCFA and the reduced richness of microbiota suggest that OS delays bacterial diversification. This is similar to what is supposed to happen in breast-fed babies compared with babies fed with unsupplemented formula [81]. The similarity is quite surprising in that the chemical nature of the constituent monomers and the pattern of glycoside linkages in different OS products are expected to influence the ability of individual bacteria to grow on them (see [31,82] for reviews). However, our results are consistent with Harris et al.’s findings [83] that the orientation of glycoside linkage is not a main driver of the SCFA production profile. When this chemical difference could act, it would primarily modulate the proportion of butyrate, an SCFA weakly produced in our immature animals. In addition, they also agree with the similarities of microbiotal impacts reported between βGOS and FOS on the one hand [30], and between αGOS and βGOS on the other [44].

Thus, our study confirms the prebiotic character of αGOS and, in addition, extends the well-known activity of FOS and GOS/In as secretagogues of satiety enteropeptides to this new prebiotic, a finding which is in accordance with the satietogenic effect described in humans [67].

## 5. Conclusions

In conclusion, our study depicts that the ability of the OS to modulate EECs as previously described in adults also operates in the neonatal period, despite the immaturity of the microbiota at this time. This observation therefore calls into question the nature of the mediators actually involved, as supposed so far. In addition, our in-depth study of the impacts of the OS on the genes regulating the differentiation of EEC precursors queries the current understanding of the ontogenesis of these cells.

Finally, our results do not demonstrate any programming impact of OS either on EECs and food consumption or on the constitution of the adult microbiota. If this holds true for humans, it is reassuring since this study concerns types and dosages of OS mimicking some of those commonly prescribed in formula for toddlers.

## Figures and Tables

**Figure 1 nutrients-11-01967-f001:**
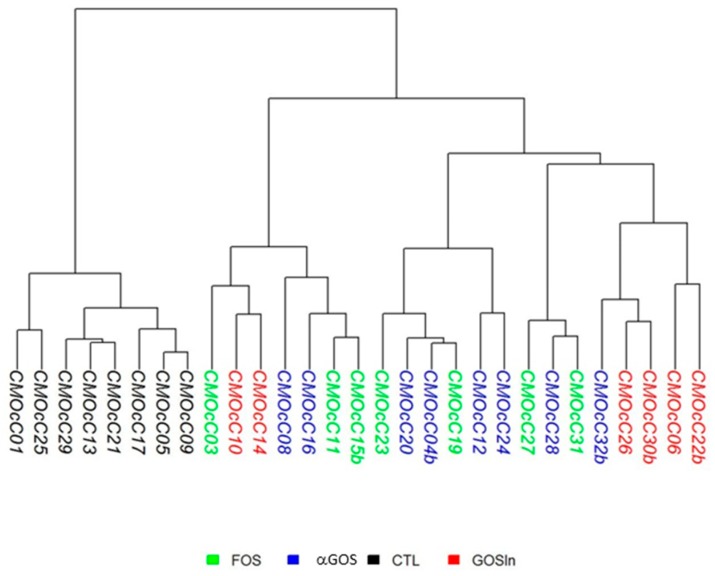
Hierarchical clustering based on the Ward’s method of phylogenetically informed distance matrix computed using the weighted UniFrac metric for cecocolonic contents collected at postnatal day (PND) 14/15 (*n* = 6 to 8 per group).

**Figure 2 nutrients-11-01967-f002:**
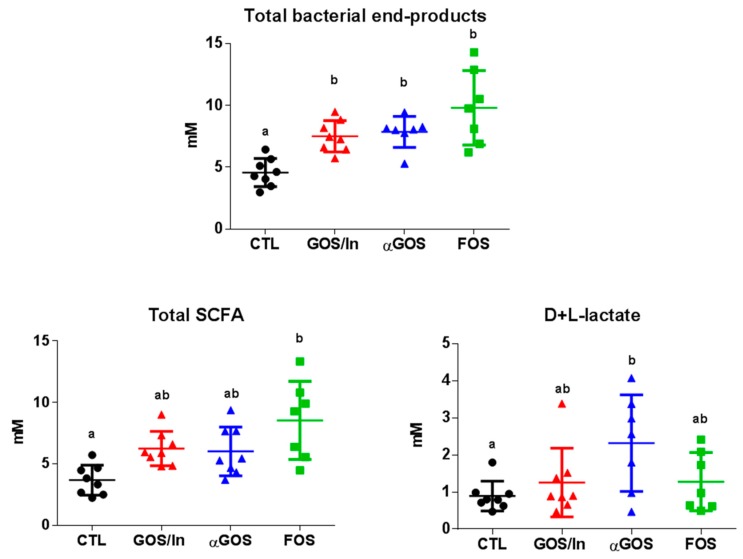
Cecocolonic concentrations of fermentation end-products. Individual, mean and SD values are plotted (*n* = 7 to 8 per group). Different letters indicated significant difference (*p* < 0.05) between groups.

**Figure 3 nutrients-11-01967-f003:**
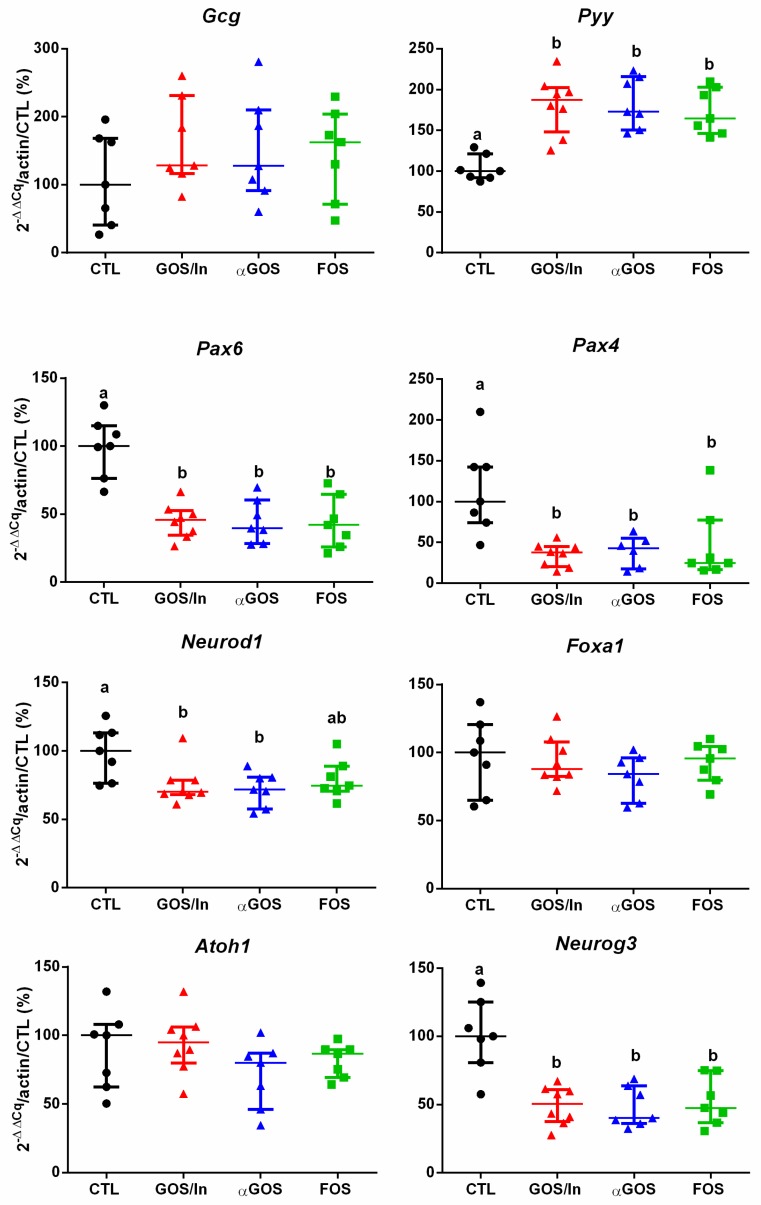
Relative expression of genes implied in the endocrine lineage and in L-cells differentiation in the ileum. Different letters indicate significant difference between groups (*p* < 0.05). Data are fold-change expressed in % of CTL group. Individual values, median with interquartile range are plotted (*n* = 7 to 8 per group).

**Figure 4 nutrients-11-01967-f004:**
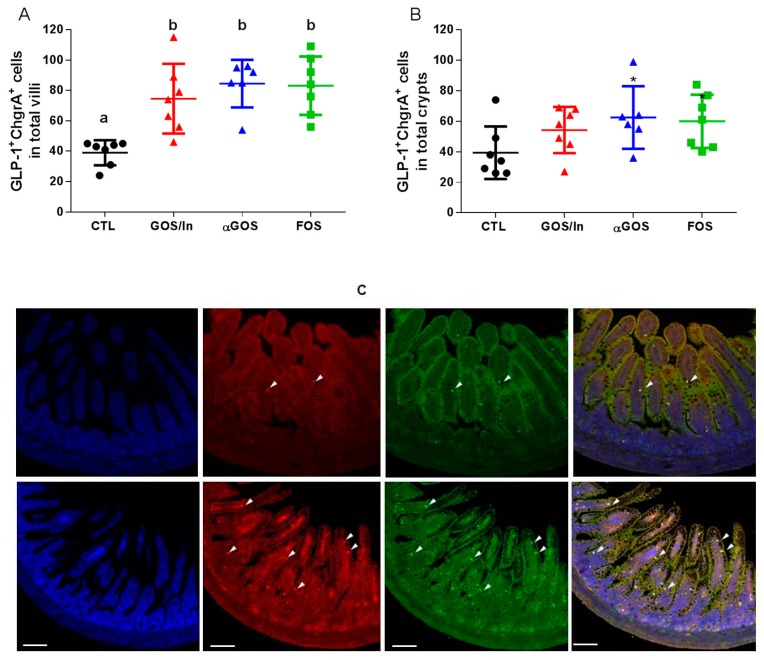
Effect of oligosaccharides (OS) supplementation on the density of GLP-1 cells in ileum: (**A**) in villi (**B**) in crypts. Different letters indicate significant differences among groups (*p* < 0.05); Individual, mean and SD values are plotted (*n* = 6 to 7 per group). (**C**) Representative images of immunofluorescence in ileal sections from a control solution (CTL) (top) and αGOS groups (down), arrows indicate positive fluorescence in cells: blue (DAPI, nuclei staining), red (GLP-1 cells), green (ChrgA cells) and merge (GLP-1/ChrgrA cells). Bars indicate 100 µm.

**Figure 5 nutrients-11-01967-f005:**
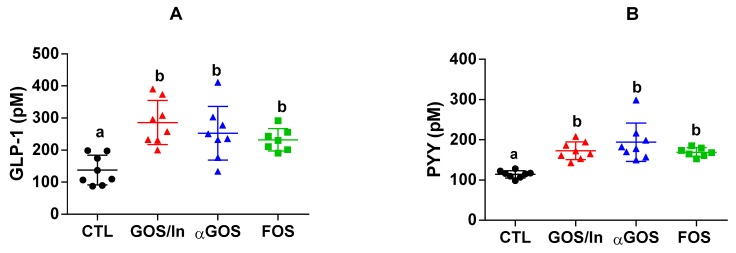
Plasma concentration of (**A**) Total GLP-1; (**B**) Total PYY at PND 14/15. Different letters indicate significant differences among groups (*p* < 0.05). Individual, mean and SD values are plotted (*n* = 7 to 8 per group).

**Figure 6 nutrients-11-01967-f006:**
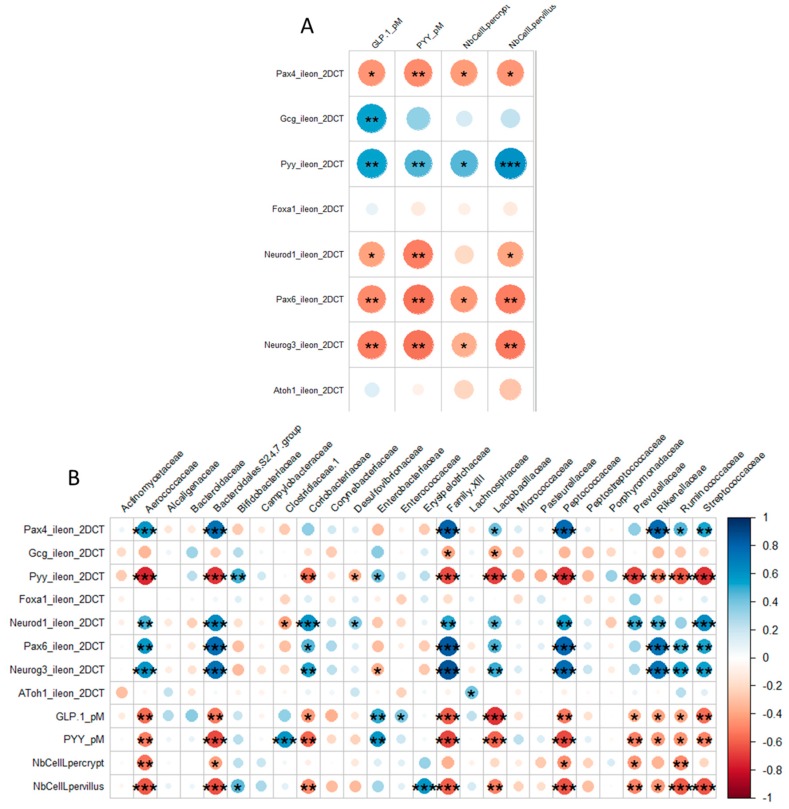
Correlograms within EEC descriptors (**A**) or between these descriptors and the relative abundances of main bacterial families (**B**). Positive correlations are displayed in blue and negative correlations in red. The intensity of the color and the size of the circles are proportional to the correlation coefficients. Asterisks indicate the level of significance (*, *p* < 0.05; **, *p* < 0.01; ***, *p* < 0.001). On the right of the correlogram, the color legend shows the correspondence between correlation coefficients and colors.

**Figure 7 nutrients-11-01967-f007:**
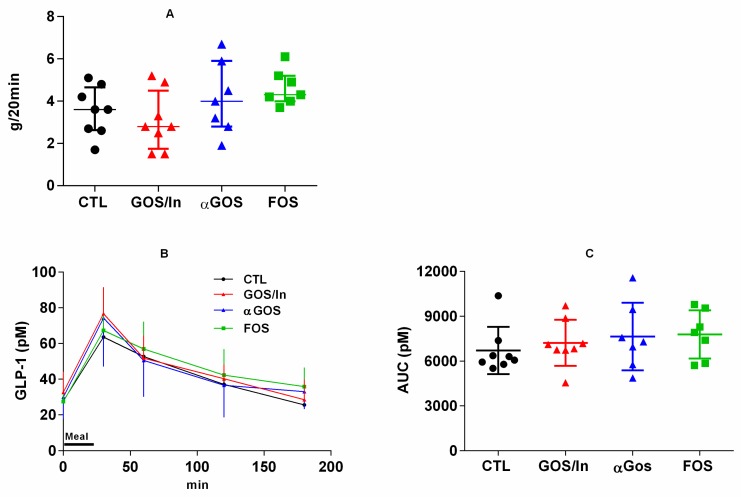
Fasting-refeeding test. (**A**) Food intake measured during refeeding (20 min-meal); (**B**) Plasma concentration of total GLP-1 measured during the 3h-kinetic follow-up (means ± SD); (**C**) Total amount of GLP-1 secreted during the 0-180min period expressed as AUC. Individuals, means and SD are plotted. (*n* = 7 to 8 per groups).

**Figure 8 nutrients-11-01967-f008:**
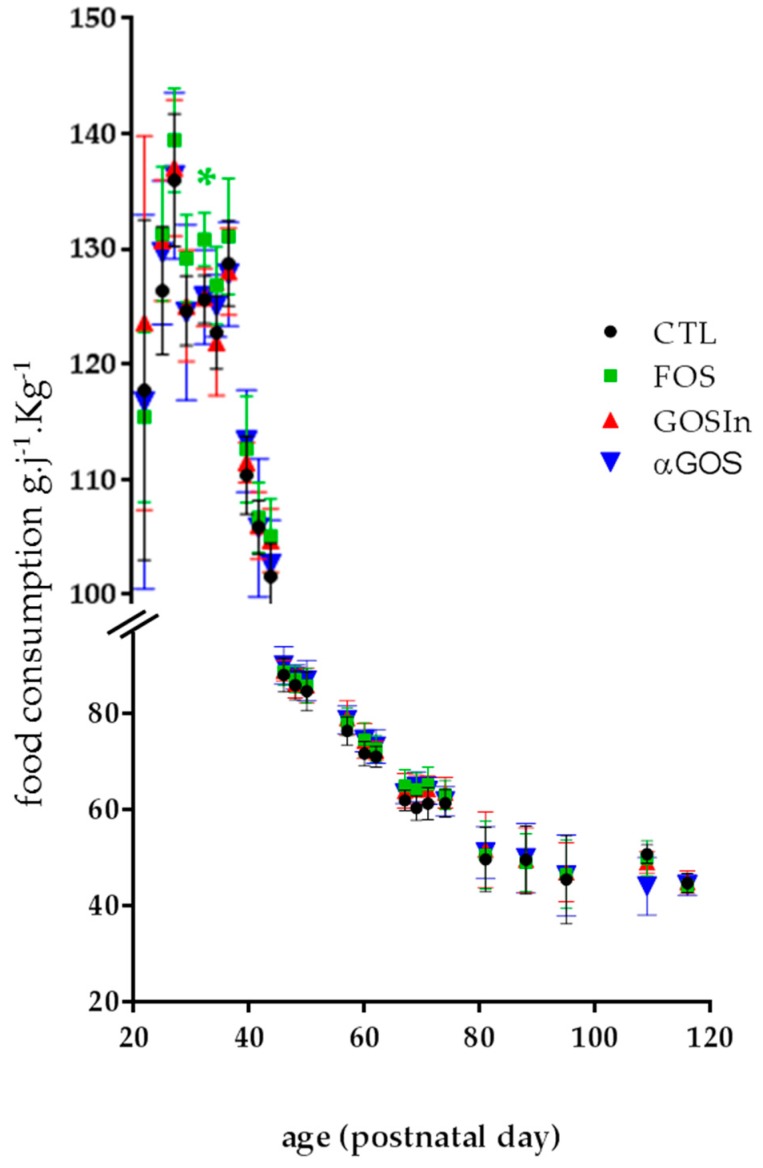
Daily consumption of food in the post-weaning stage, expressed as kilograms of bodyweight. The asterisk indicates a significant difference between FOS and CTL groups (*p* < 0.05). Data are means ± SD (*n* = 7 to 8 by group and day).

**Figure 9 nutrients-11-01967-f009:**
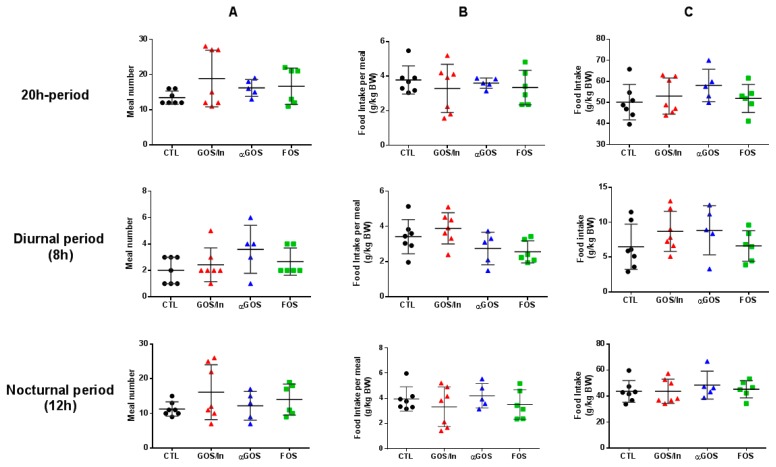
Feeding patterns illustrated by (**A**) Meal number; (**B**) Food intake per meal; (**C**) Food intake during the considered period analyzed in physiological cages at PND 75–100 (*n* = 5–7 per group). BW, bodyweight. Individuals, means and SD are plotted (*n* = 5 to 7 per groups).

**Figure 10 nutrients-11-01967-f010:**
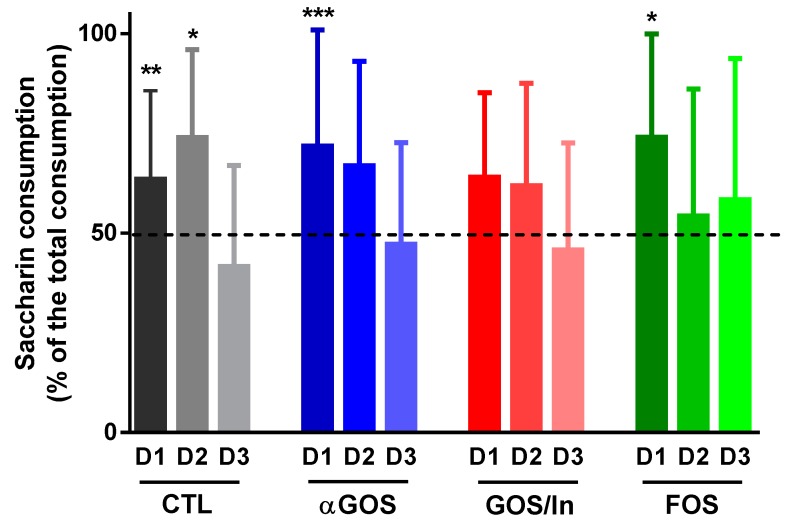
Preference for sweet taste. Data are means ± SD (*n* = 7 to 8). Asterisks represent significant preference as compared with no preference (i.e., 50%, dotted line): *, *p* < 0.05; **, *p* < 0.01; ***, *p* < 0.001.

**Figure 11 nutrients-11-01967-f011:**
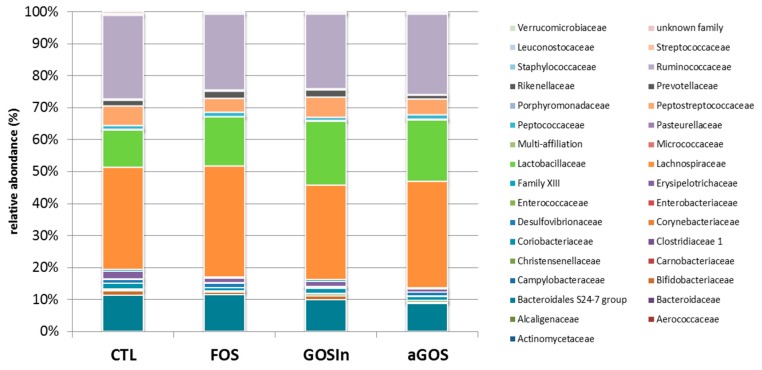
Impact of postnatal OS-supplementation on cecal microbiota composition at PND124/126: families distribution expressed as the average of cumulated relative abundances (*n* = 7 to 8 per group).

**Table 1 nutrients-11-01967-t001:** Composition of solutions administered by gavage to pups from PND5 to PND14/15 (g·mL^−1^).

	CTL	FOS	GOS/In	αGOS
GOS syrup (VivinalGOS, FrieslandCampina Domo, LE Amersfoort, The Netherlands)			0.65	
Inulin powder (Raftiline HP, BENEO-Orafti S.A., Tienen, Belgium)			0.03	
FOS powder (Beneo P95, BENEO-Orafti S.A., Tienen, Belgium)		0.34		
αGOS powder (Olygose, Venette, France)				0.32
α-Lactose monohydrate (L3625, Sigma-Aldrich, St. Quentin Fallavier, France)	0.096	0.096		0.096
D(+)-glucose monohydrate (108342, Merck Santé SAS, Fontenay sous Bois, France)	0.087	0.082		0.087
D(+)-galactose monohydrate (104058, Merck Santé SAS, Fontenay sous Bois, France)	0.005	0.005		0.004
D(−)-fructose (F0127, Sigma-Aldrich, St. Quentin Fallavier, France)	0.015		0.015	0.015
Saccharose (S9378, Sigma-Aldrich, St. Quentin Fallavier, France)	0.002		0.002	0.002
Total oligosaccharides ^§^		0.30	0.30	0.30
Total digestible sugars ^§ ‼^	0.20	0.20	0.20	0.20

CTL, control; FOS, fructo-oligosaccharides; 93.2% dry matter composed of 90.4% oligomers and 6.6% monomers, providing 0.015 g·mL^−1^ of fructose, 0.005 g·mL^−1^ of glucose and 0.002 g·mL^−1^ of saccharose; GOS/In, mix (9:1) of galacto-oligosaccharides and long chain fructo-oligosaccharides (In, inuline). For GOS: 75% dry matter composed of 59% oligomers and 41% monomers; for inulin: 97% dry matter composed of 99.5% oligomers, the mix was providing 0.095 g·mL^−1^ of lactose, 0.086 g·mL^−1^ of glucose and 0.005 g·mL^−1^ of galactose; αGOS: alpha galacto-oligosaccharides (95.9% dry matter composed of 99.4% oligomers, providing 0.001 g·mL^−1^ of galactose. ^!!, §^ These calculations take into account the dry matter of the components, their purity, and the amount of digestible sugars they contain.

**Table 2 nutrients-11-01967-t002:** Bodyweight gain (g) during lactation.

Treatment	BW Gain PND0-14	BW Gain PND0-20
CTL	30.4 ± 4.2 ^1^	50.5 ± 6.0
FOS	28.0 ± 3.4	45.7 ± 4.6
GOS/In	29.4 ± 3.3	49.6 ± 5.8
αGOS	27.9 ± 2.7	46.8 ± 5.1

^1^ Data are means ± SD collected from the total effective of rats (*n* = 15–16 per group during PND0-14 and *n* = 8 during PND0-20). BW, bodyweight.

**Table 3 nutrients-11-01967-t003:** Relative abundances (%) for families with abundances > 0.01% at PND14/15 according to the postnatal OS supplementation.

Family	CTL	FOS	GOSIn	αGOS
Actinomycetaceae	0.095 ± 0.084 ^1^	0.028 ± 0.032	0.076 ± 0.056	0.068 ± 0.058
Aerococcaceae	0.086 ± 0.029 ^a,2^	0.011 ± 0.011^b^	0.015 ± 0.012 ^b^	0.020 ± 0.019 ^b^
Alcaligenaceae	0.020 ± 0.041	0.030 ± 0.047	0.378 ± 0.576	0.036 ± 0.053
Bacteroidaceae	2.352 ± 0.991	6.510 ± 10.047	6.837 ± 5.729	3.719 ± 5.492
Bacteroidales.S24.7 group	6.812 ± 2.953 ^a^	0.053 ± 0.080 ^b^	0.098 ± 0.094 ^b^	0.084 ± 0.103 ^b^
Bifidobacteriaceae	0.624 ± 0.45 ^a^	17.188 ± 12.735 ^b^	7.894 ± 7.947 ^ab^	13.577 ± 10.631 ^b^
Campylobacteraceae	0.009 ± 0.024	0.093 ± 0.220	0.066 ± 0.151	0.294 ± 0.546
Clostridiaceae.1	0.273 ± 0.146	2.413 ± 3.231	5.509 ± 8.749	5.044 ± 4.581
Coriobacteriaceae	0.108 ± 0.039 ^a^	0.039 ± 0.034 ^b^	0.036 ± 0.041 ^b^	0.023 ± 0.018 ^b^
Corynebacteriaceae	0.032 ± 0.023	0.007 ± 0.011	0.020 ± 0.030	0.012 ± 0.023
Desulfovibrionaceae	0.098 ± 0.182	0.000 ± 0.000	0.003 ± 0.008	0.006 ± 0.014
Enterobacteriaceae	13.86 ± 5.97 ^a^	23.48 ± 12.23 ^ab^	19.51 ± 6.69 ^b^	33.42 ± 11.99 ^b^
Enterococcaceae	0.435 ± 0.707	0.145 ± 0.203	2.892 ± 6.293	0.542 ± 0.771
Erysipelotrichaceae	0.682 ± 0.387	4.080 ± 3.988	3.774 ± 4.950	2.766 ± 3.002
Family.XIII	0.062 ± 0.030 ^a^	0.004 ± 0.008 ^b^	0.000 ± 0.000 ^b^	0.001 ± 0.003 ^b^
Lachnospiraceae	6.327 ± 2.300 ^a^	9.787 ± 6.180 ^ab^	15.298 ± 9.544 ^b^	4.962 ± 4.587 ^b^
Lactobacillaceae	57.47 ± 8.72 ^a^	28.74 ± 10.84 ^b^	24.47 ± 5.71 ^b^	31.13 ± 11.24 ^b^
Micrococcaceae	0.140 ± 0.064	0.075 ± 0.074	0.071 ± 0.053	0.110 ± 0.105
Pasteurellaceae	0.582 ± 0.581	0.236 ± 0.235	0.456 ± 0.297	0.394 ± 0.446
Peptococcaceae	0.396 ± 0.182 ^a^	0.006 ± 0.015 ^b^	0.015 ± 0.019 ^b^	0.007 ± 0.021 ^b^
Peptostreptococcaceae	0.747 ± 0.485	0.471 ± 0.262	0.543 ± 0.108	0.640 ± 0.379
Porphyromonadaceae	1.242 ± 1.153	5.924 ± 9.747	9.826 ± 15.228	2.055 ± 5.475
Prevotellaceae	2.136 ± 1.540 ^a^	0.014 ± 0.016 ^b^	0.011 ± 0.018 ^b^	0.028 ± 0.060 ^b^
Rikenellaceae	0.034 ± 0.039 ^a^	0.001 ± 0.004 ^b^	0.000 ± 0.000 ^b^	0.001 ± 0.003 ^b^
Ruminococcaceae	3.242 ± 0.743 ^a^	0.135 ± 0.147 ^b^	1.610 ± 2.622 ^b^	0.406 ± 0.665 ^b^
Streptococcaceae	2.118 ± 0.620 ^a^	0.510 ± 0.316 ^b^	0.586 ± 0.156 ^b^	0.643 ± 0.450 ^b^

^1^ Data are means ± SD (*n* = 6 to 8 per group). ^2^ Within a row, values followed by different letters (a,b,ab) differ significantly (*p* < 0.05).

**Table 4 nutrients-11-01967-t004:** Concentration (mM) of major short chain fatty acids (SCFA) in cecocolonic contents at PND 14/15.

Treatment	Acetate	Propionate	Butyrate	pH
CTL	3.17 ± 1.05 ^1,a,2^	0.39 ± 0.16	0.07 ± 0.04	6.9 ±0.3 ^a^
GOS/In	5.82 ± 1.32 ^b^	0.33 ± 0.21	0.10 ± 0.09	6.3 ±0.2 ^b^
αGOS	5.69 ± 1.77 ^ab^	0.28 ± 0.27	0.05 ± 0.00	6.1 ±0.2 ^b^
FOS	8.00 ± 2.94 ^b^	0.47 ± 0.37	0.06 ± 0.04	6.2 ±0.2 ^b^

^1^ Data are means ± SD (*n* = 7 to 8 per group). ^2^ Within columns, values followed by different letters (a,b,ab) differ significantly at *p* < 0.05.

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
