# Peer review of "Neonatal Consumption of Oligosaccharides Greatly Increases L-Cell Density without Significant Consequence for Adult Eating Behavior"

_nutrients, 2019, doi:10.3390/nu11091967_

Round 1
Reviewer 1 Report
This is a very interesting paper.
Your results agree with the results of two our papers, that are "in vivo studies", were we demonstrated that external interventions with probiotics didn't determine any differences in microbiota composition (I suggest to cite them in paragraph 3.3.2:
Effectiveness and Safety of a Probiotic-Mixture for the Treatment of Infantile Colic: A Double-Blind, Randomized, Placebo-Controlled Clinical Trial with Fecal Real-Time PCR and NMR-Based Metabolomics Analysis.
Baldassarre ME, Di Mauro A, Tafuri S, Rizzo V, Gallone MS, Mastromarino P, Capobianco D, Laghi L, Zhu C, Capozza M, Laforgia N.
Nutrients. 2018 Feb 10;10(2). pii: E195. doi: 10.3390/nu10020195
Administration of a Multi-Strain Probiotic Product to Women in the Perinatal Period Differentially Affects the Breast Milk Cytokine Profile and May Have Beneficial Effects on Neonatal Gastrointestinal Functional Symptoms. A Randomized Clinical Trial.
Baldassarre ME, Di Mauro A, Mastromarino P, Fanelli M, Martinelli D, Urbano F, Capobianco D, Laforgia N.
Nutrients. 2016 Oct 27;8(11). pii: E677)
I think that the authors must better specified in methods how they have chosen the composition and concentration of the different solutions to administer to pup rats.
Then I think that the paper need a good English revision, i.e.
Line 37: Healthy problems
Line 86: we aimed to evaluate
Line 89: we choose
Line 113: three times a week
etc.
Author Response
"Please see the attachment."

Reviewer 2 Report
In this study, the authors describe the role of the administration of different prebiotics between P5-15 on different biochemical measures on P15 or biochemical and eating measures on adult. The manuscript brings up some interest, I believe the experimental design and the content of the manuscript could be better cared. I don’t think the title is appropriate as eating behaviour was barely explored in this paper. Also, the study cannot proof that the composition used here have an effect of eating behaviour at all (eg. by short-term effect). The structure of the figures must be largely improved. Whenever possible, the experiments done in neonates and adults that are similar should be placed together for visual comparisons. I also could not find the supplement on the website, which I believe it is a technical issue from the publisher. Other comments below.
Line 66: the authors mention dysbiosis to refer to a microbial imbalance in the gut, which is correct. However, the study looks a beneficial change in the gut microbiota where the term dysbiosis could be misleading since it points to both direction (it can be a bacterial overgrowth but also an underrepresentation of some bacterial colonies). The authors could be more specific demonstrating when and how the gut microbial imbalances may affect eating disorders. Also, eating disorders is a very unspecific term for this study because the authors are looking for a beneficial effect of the eating behaviour in healthy subjects (eg. increase feeding). For the introduction, the authors could also exemplify better which types of eating disorders exist, as like dysbiosis, eating disorders can represent extremes of eating behaviour (eg. bulimia – people eat large amount of food; anorexia – people are not eating enough food).
Line 68: could the authors cite an experimental study demonstrating that GF differ from conventional animals in feeding behaviour? I could not find in the review that was cited here.
Line 68: the authors assume that prebiotics are the only microbiota-disrupting agents that can affect feeding behaviour. Can the authors exclude the participation of probiotics and other microbiota-disrupting agents (eg. Stress, exercise, diet...) on feeding behaviour?
This sentence seems confusing: “as well as the ability of the microbiota [28] and of certain prebiotics [29,30] to modulate some behaviors in adults mice can be invoked”. The authors suggest that prebiotics are an intervention (treatment) for the microbiota, which is correct, but the way the sentence is written suggest that the microbiota can be a therapy as well. It can be the case in situations like FMT or co-housing, but I don’t it is what the authors want to express here. Moreover, other types of therapy can intervene the gut microbiota and deserve attention.
L117: the word slaughtering can sound a bit hash for general audience. Please use a more suitable wording for publication (eg. cull, sacrifice…)
L156: did the authors control for the crumbs that fall from the grid to the floor of the homecage? By experience this can be of significant amount.
L233: why number of animals used for 16S are smaller? Also, why number of animals vary for experiment to experiment?
L278: it is not clear what is bacterial end-products and how it differs from total SCFA.
L282: Where is written table 3, it is actually table 4.
Figure 3 and 5: if the values are relative to control, shouldn’t the control be 100% in all graphs?
Figure 4: please provide an illustrative image for the immuno.
L373: the feeding behaviour is one the main goal of this study (also included in the title). The authors should show the data of the feeding behaviour even though it is not significant.
Figure 10: since the GOS/In group was the only group not affected by the saccharin challenge, and the GOS/In solution present a distinct composition of the excipients compared to CTRL and others experimental groups (eg. absence of lactose, glucose and galactose). Can the authors exclude an effect of these components on the behavioural output observed in this experiment? How?
Table 3/Figure 11: why the authors decided to use distinct methods to show the same type of data. Personally, a graphic looks better.
Figure 8: perhaps I am missing something. Is it the food intake way higher during the early life?
L408: why the experiment was designed to cover 20h of the day instead of 24h? What happened with the 4h left?
L483: perhaps the eating behaviour is also not affected in neonates (Figure 8 can illustrate this) and prebiotics doesn’t have an effect at eating behaviour at least in the conditions studied here.
L489-490: this sentence can be misinterpreted as the prebiotics used here can also affect the microbiota in adults (see Burokas et al paper) when given to adults. The results of this manuscript only observed that the prebiotics doesn’t have a long term effect on rats and that majority of the effects observed are due to short term action.
L493-95: I don’t agree with the affirmations of this sentence. Acetate seems also to be affected by prebiotics and not only propionate and butyrate (see again Burokas et al paper).
L497-500: Again, this conclusion may not be true. One can argument that administration of prebiotics in adults can also can have the same effect from the ones observed here. I believe that an adult-treated group is missing here to have such conclusion.
L537: when CCK was investigated (neonate or adults)?
L566: could individual differences also contribute with these disparities?
Could the authors give the references for the studies mentioned here? “probably depending on the mode (orogastric, intraperitoneal, intracerebroventricular, colonic delivery via fermentable fibers, etc.) and duration (acute vs chronic) of SCFA or SCFA precursors administration”
L593: where in the rat brainstem c-fos was analysed?
L593-95: can we assume that postnatal modulation of the gut microbiota by OS is stress? If yes, protein restriction and maternal deprivation would far beyond the OS treatment to be comparable.
Author Response
"Please see the attachment."

Reviewer 3 Report
It is a paper with a correct approach and suitable methodology.
The results in some aspect are unexpected. It might be thought that the
differentiated activity of the EEC can have a long-term effect.
In the first days of life there is an increase in weight in the control
group, the body composition could be studied in future work. The
difference in weight can be due to the accumulation of fats or by
increased muscle or bony mass.
Author Response
We thank the reviewer for this comment and will consider her/his advice in our future works.